Journal of
open psychology data

# Data from the Panel Study 'Refugees in the German Educational System (ReGES)'

COLLECTION:
DATA FOR
PSYCHOLOGICAL
RESEARCH IN THE
EDUCATIONAL FIELD

DATA PAPER

JUTTA VON MAURICE

GISELA WILL [iD]

*Author affiliations can be found in the back matter of this article

]u[ubiquity press

## ABSTRACT

The study 'Refugees in the German Educational System' is a two-cohort panel addressing the integration of refugee children and adolescents into the German educational system. Data collection followed a multi-informant perspective as well as a multi-mode approach. It started at Wave 1 in January 2018 with a sample of 2,405 refugee children and 2,415 refugee adolescents. Participants were followed over seven survey waves for more than two years. ReGES data is stored at the Research Data Center of the Leibniz Institute for Educational Trajectories and is open for use for to scientific community without costs or any embargo.

CORRESPONDING AUTHOR:
**Jutta von Maurice**

Leibniz Institute for
Educational Trajectories, DE

jutta.von-maurice@lifbi.de

KEYWORDS:
refugees; preschool; school; integration; education

TO CITE THIS ARTICLE:
von Maurice, J., & Will, G. (2023). Data from the Panel Study 'Refugees in the German Educational System (ReGES)'. *Journal of Open Psychology Data,* 11: 1, pp. 1–15. DOI: https://doi.org/10.5334/jopd.77

# (1) BACKGROUND

In the 2010s—with a peak in the middle of the decade—many refugees fled to the European Union and especially to Germany. A total of 2.1 million people applied for asylum in Germany from 2010 to 2019 and 788,053 of these were minors (37.5%). Figure 1 shows a clear peak in first-time applications for asylum in 2015 and 2016.

The panel study 'Refugees in the German Educational System (ReGES)' was funded by the German Federal Ministry of Education and Research and located at the Leibniz Institute for Educational Trajectories (LIfBi). It addressed the refugee population that applied for asylum in Germany in the mid-2010s with a clear focus on minors and their families. Viewing education as a key contributor to the integration of refugee minors, ReGES focused on factors that may foster or hinder integration in the educational system (Will et al., 2018). Classical factors from migration research (standardized measures of language proficiency in the host country language, ethnic networks, residence status, length of stay in Germany) were also included as well as refugee-specific factors (such as flight history, indicators of traumatization, type of accommodation). In addition, a number of factors were recorded that previous educational research had shown to be relevant for educational success (e.g. basic cognitive functioning, socio-economic background of parents, parental supportive behaviour).

ReGES was conceptualized as a two-cohort panel study focusing on selected transitions within the educational system (for a brief overview of the German school system, see Secretariat of the Standing Conference of the Ministers of Education and Cultural Affairs of the Länder in the Federal Republic of Germany, 2019). The two cohorts were:

- *Refugee Cohort 1* containing 2,405 children aged 4 years and above but not yet attending school at Wave 1. The focus was on preschool attendance or other forms of child care as well as on the transition to school. Due to the age of the children, data was collected primarily from the parents.
- *Refugee Cohort 2* containing 2,415 adolescents aged 14 to 16 years at the time of sampling who were still attending lower secondary school at Wave 1. The focus was on early school experiences, transitions within the general school system, and transitions to the vocational educational system or tertiary education. The main respondents within this cohort were the adolescents themselves.

The sample was drawn in five German Federal States. Following the longitudinal design, participants were surveyed in seven panel waves covering an observation period from spring 2018 till autumn 2020. Taking a multi-informant perspective helped to gain an in-depth understanding of the family as well as relevant educational institutions and living constellations that support or hinder integration. The design of both ReGES cohorts followed a clear multi-method approach and included personal interview settings as well as telephone and online interviews as the main survey modes (see Section 2.1). This approach was used because no valid information was available on the refugee group under study with regard to response rates in different survey modes. ReGES aimed to acquire survey methodological knowledge on the most appropriate way to address refugees while also taking survey costs into account.

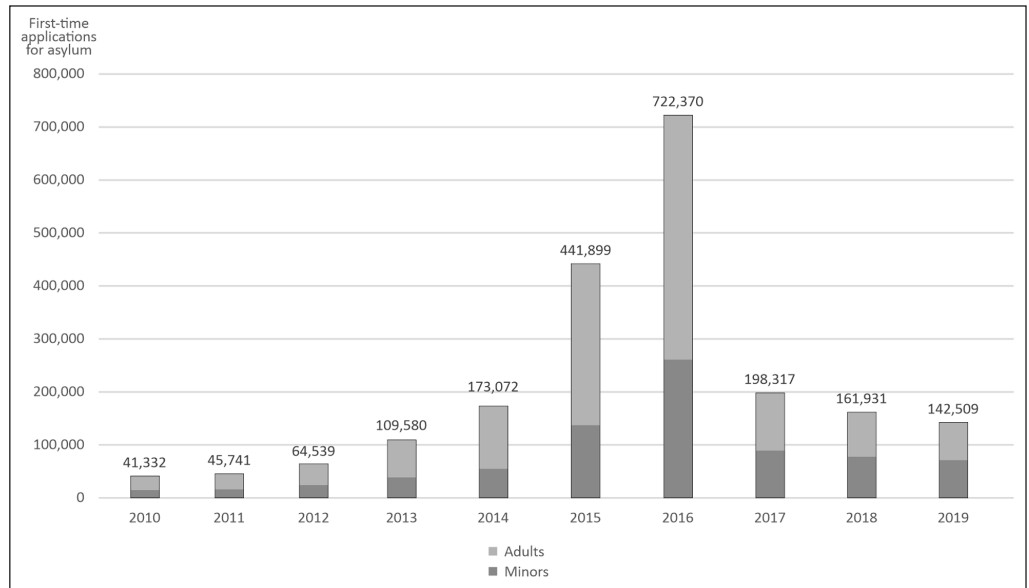

**Figure 1** First-time applications for asylum in Germany 2010–2019 (data taken from the reports of the Federal Office for Migration and Refugees [*Bundesamt für Migration und Flüchtlinge*] from 2011 till 2020).

## (2) METHODS

ReGES sampled two cohorts of refugees in Germany: 2,405 children of preschool age and 2,415 adolescents in lower secondary schools. Both cohorts were surveyed across the seven survey waves from spring 2018 till autumn 2020.

### 2.1 STUDY DESIGN

Figure 2 presents the seven panel waves in ReGES. It shows primarily in which educational stages the refugee children and adolescents under study are ideally in at these seven waves and which central transitions in the German educational system they have to master during the progress of the panel. The upper half of Figure 2 focuses on Refugee Cohort 1, the lower half on Refugee Cohort 2. Figure 2 also shows in which waves data of children and adolescents (as the main targets of our research questions) is directly collected but also the measurement of additional information given from parents, educational professionals (preschool and school teachers and heads) and administration staff—following a clear multi-informant design.

Refugee Cohort 1 focused on preschool-aged children with the parents being the main survey respondents (but children being included for direct competence measurement). Refugee Cohort 2 shed light on the situation of refugees in secondary education with adolescents being the main survey respondents (but parents being included for background information in the first wave). At the centre of the data collection were computer-assisted personal interviews (CAPI) and computer-assisted self-interviews (CASI) for parents and adolescents conducted in the refugees' homes (Waves 1, 4, and 7). Within these home visits, technology-based

competence tests (TBT) were carried out with children and adolescents (see also Section 2.5 for more details concerning this design aspect and its technological implementation). Additional data stemmed from computer-assisted telephone interviews (CATI) as well as computer-assisted web interviews (CAWI) for adolescents and parents. Detailed information on the basic design parameters of Refugee Cohort 1 is given in Table 1 and of Refugee Cohort 2 in Table 2.

As Table 1 and Table 2 show with respect to the valid realized sample participation rates vary substantially with the interview mode. While the parents and adolescents can best be reached in personal interviews (Waves 1, 4, and 7), followed by telephone interviews (Wave 3), online surveys (Waves, 2, 5, and 6) are the least effective. We also see different selectivity patterns, depending on the survey mode used (see Heinritz, & Will, 2021): The selective participation by education—which is particularly harmful for education-related studies—is least pronounced in the personal interview setting.

In addition to the refugee children, adolescents, and their parents, relevant context persons were integrated into the design. For a detailed understanding of the processes in and effects of educational institutions, preschool teachers and principals and—after school enrolment—school teachers and principals were included in Refugee Cohort 1. In Refugee Cohort 2, school teachers and principals were included; in later waves, also staff in vocational schools. In both cohorts, staff in the collective accommodation and municipalities were surveyed in the first wave. Information from these context persons was collected via paper-and-pencil questionnaires (PAPI) that were distributed via regular mail. By taking this broad multi-informant perspective, the ReGES datasets allow an in-depth understanding of processes relevant for refugee integration.

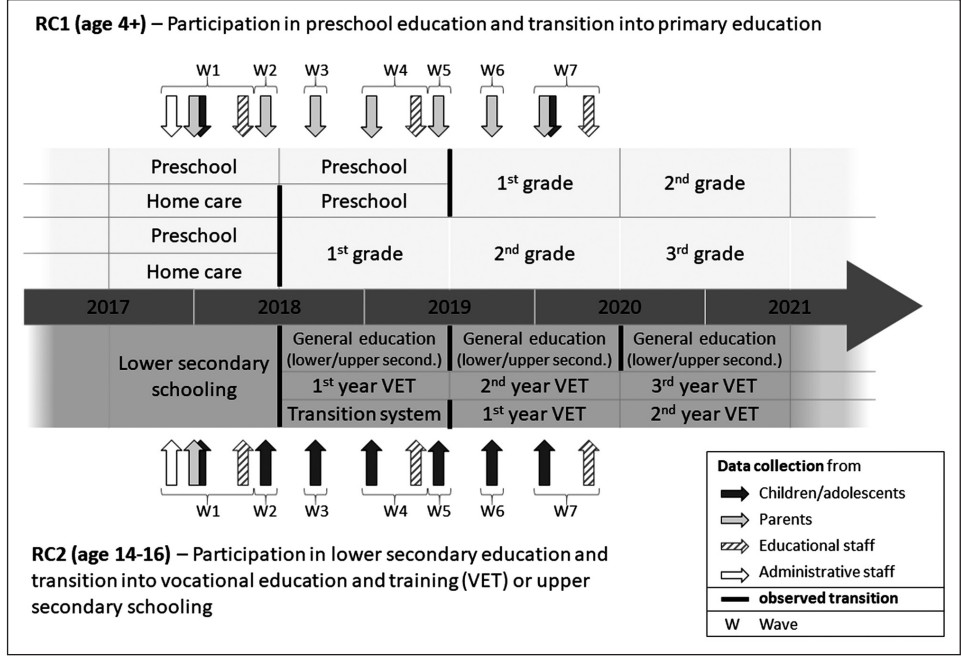

**Figure 2** ReGES study design (taken from Will et al., 2021).

| WAVE | MODE AND INFORMATION ON DURATION (*M/SD* IN MINUTES) | LANGUAGES | GROSS SAMPLE | VALID REALIZED SAMPLE | FIELD PHASE | INCENTIVE |
|---|---|---|---|---|---|---|
| Wave 1[1,2] | Parents: CAPI/CASI (*M* = 53.6; *SD* = 29.0)[3]; Children: TBT (*M* = 23.7; *SD* = 11.5) | CAPI/CASI: Arabic, English, Farsi, French, German, Kurmanji, Pashto, Tigrinya; TBT (instruction only): Arabic, English, German, Kurmanji | Gross sample: 4,680 addresses; screening; Response Rate 2 = 48.3%; Cooperation Rate 4 = 78.6% | Parents: 2,075 (of 2,405 children); Children: 2,405 (1,498 with complete competence measurement) | 29/01/2018–30/06/2018[4] | Parents: 20 Euro drugstore voucher (post-paid); Children: Painting book and pens (during interview) |
| Wave 2[5] | Parents: App-based CAWI (short) (*M* = 1.9; *SD* = 1.7) | Arabic, English, Farsi, French, German, Kurmanji, Pashto, Tigrinya | Parents: 1,989 (of 2,312 children) | Parents: 287 (of 323 children) | 29/01/2018–03/08/2018[4] | Parents: 5 Euro drugstore voucher for app download (prepaid) |
| Wave 3 | Parents: CATI (*M* = 45.3; *SD* = 15.6)[3] | Arabic, English, German, Kurmanji | Parents: 1,935 (of 2,239 children) | Parents: 791 (of 887 children) | 27/08/2018–24/11/2018 | Parents: 10 Euro drugstore voucher (post-paid) |
| Wave 4[1] | Parents: CAPI/CASI (*M* = 53.1; *SD* = 22.4)[3] | Arabic, English, German, Kurmanji | Parents: 1,915 (of 2,216 children) | Parents: 1,484 (of 1,703 children) | 04/02/2019–31/05/2019[4] | Parents: 20 Euro drugstore voucher (post-paid); Children: Painting book and pens (during interview) |
| Wave 5[5] | Parents: Online (invitation via App, mail or e-mail; short) (*M* = 7.6; *SD* = 3.6) | Arabic, English, Farsi, French, German, Kurmanji, Pashto, Tigrinya | Parents: 1,929 (of 2,228 children) | Parents: 349 (of 401 children) | 15/04/2019–30/08/2019[4] | Parents: 5 Euro drugstore voucher (post-paid) |
| Wave 6 | Parents: Online (long) (*M* = 25.2; *SD* = 9.2) | Arabic, English, German, Kurmanji | Parents: 1,852 (of 2,142 children) | Parents: 446 (of 522 children) | 04/10/2019–17/11/2019 | Parents: 10 Euro drugstore voucher (post-paid) |
| Wave 7[1,6] | Parents: CAPI/CASI/CAPI-by-phone (*M* = 59.7; *SD* = 21.1)[3]; Children: TBT (*M* = 31.2; *SD* = 8.2) | CAPI/CASI/CAPI-by-phone/TBT (instruction only): Arabic, English, German, Kurmanji | Parents: 1,848 (of 2,136 children) | Parents: 1,053 (of 1,199 children); Children: 790 (with complete competence measurement) | 10/02/2020–05/09/2020 | Parents: 20 Euro drugstore voucher (post-paid); Children: puzzle block (after competence test) |

**Table 1** Methodological fact sheet for ReGES sample: Refugee Cohort 1.

*Note:* Data taken from several field reports (Ruland, Sandbrink, Cohrs, & Hess, 2019, 2020; Ruland, Sandbrink, Cohrs, Weiß, & Hess, 2019; Ruland, Sandbrink, & Hess, 2019a, 2019b; Ruland, Steinwede et al., 2019), the Scientific Use Files, and Heinritz and Will (2021). There are some minor differences between field reports and later data versions due to data cleaning procedures. The Response Rate 2 as well as the Cooperation Rate 4 were calculated according to AAPOR standards (The American Association for Public Opinion Research, 2016). This table is restricted to direct data collection with children and parents. [1]Additional data collected from educational staff in the preschools and schools children were attending at that time (PAPI); field phase deviating from parent/child survey. [2]Additional data collected from staff members in collective accommodation and municipalities (PAPI); field phase deviating from parent/child survey. [3]The information on the duration of the interview refers to the pure interview time and includes neither the screening in Wave 1 nor the contacting phases. [4]The field phases of Waves 1 and 2 as well as Waves 4 and 5 overlap because after the personal interviews the respondents were invited to the online surveys by the interviewers; thus, the respondents were able to carry out these surveys relatively soon after the personal interview. [5]Due to the larger number of languages offered in the questionnaires via App, we included some additional cases that were not eligible for the waves with reduced languages. [6]Design switch due to the Corona pandemic (because household visits had to be cancelled, TBT was no longer possible; for details, see Will et al., 2020).

| WAVE | MODE AND INFORMATION ON DURATION ($M/SD$ IN MINUTES) | LANGUAGES | GROSS SAMPLE | REALIZED SAMPLE | FIELD PHASE | INCENTIVE |
|---|---|---|---|---|---|---|
| Wave 1[1,2] | Parents: CAPI/CASI ($M = 46.8$; $SD = 25.8$)[3]; Adolescents: CASI/TBT[3] ($M = 65.5$; $SD = 32.7$) | CAP/CASI: Arabic, English, Farsi, French, German, Kurmanji, Pashto, Tigrinya; TBT (instruction only): Arabic, English, German, Kurmanji | Gross sample: 5,556 addresses screening; Response Rate 2 = 51.2%; Cooperation Rate 4 = 81.5% | Parents: 1,499 (of 1,694 adolescents); Adolescents: 2,415 (1,351 with complete competence measurement) | 29/01/2018–30/06/2018[4] | Parents: 20 Euro drugstore voucher (post-paid); Adolescents: 20 Euro drugstore voucher (post-paid) |
| Wave 2[5] | Parents/Adolescents: App-based CAWI (short) (parents: $M = 16.0$, $SD = 1.5$; adolescents: $M = 1.8$; $SD = 1.5$) | Arabic, English, Farsi, French, German, Kurmanji, Pashto, Tigrinya | Parents: 1,449 (of 1,632 adolescents); Adolescents: 2,273 | Parents: 244 (of 271 adolescents); Adolescents: 355 | 29/01/2018–03/08/2018[4] | Parents/Adolescents: 5 Euro drugstore voucher for app download (prepaid) |
| Wave 3 | Adolescents: CATI ($M = 41.7$; $SD = 10.9$)[3] | Arabic, English, German, Kurmanji | Adolescents: 2,267 | Adolescents: 935 | 27/08/2018–24/11/2018 | Adolescents: 10 Euro drugstore voucher (post-paid) |
| Wave 4[1] | Adolescents: CAPI/CASI ($M = 51.1$; $SD = 21.4$)[3] | Arabic, English, German, Kurmanji | Adolescents: 2,240 | Adolescents: 1,769 | 04/02/2019–31/05/2019[4] | Adolescents: 20 Euro drugstore voucher (post-paid) |
| Wave 5[5] | Parents/Adolescents: Online (invitation via App, mail, or e-mail; short) (parents: $M = 8.2$; $SD = 4.1$; adolescents: $M = 6.3$; $SD = 3.2$) | Arabic, English, Farsi, French, German, Kurmanji, Pashto, Tigrinya | Parents: 1,181 (of 1,249 adolescents); Adolescents: 2,190 | Parents: 138 (of 204 adolescents); Adolescents: 434 | 15/04/2019–30/08/2019[4] | Parents/Adolescents: 5 Euro drugstore voucher (post-paid) |
| Wave 6 | Adolescents: Online (long) ($M = 24.6$; $SD = 10.9$) | Arabic, English, German, Kurmanji | Adolescents: 2,166 | Adolescents: 641 | 04/10/2019–17/11/2019 | Adolescents: 10 Euro drugstore voucher (post-paid) |
| Wave 7[1,6] | Adolescents: CAPI/CASI/CAPI-by-phone/TBT ($M = 74.6$; $SD = 20.9$)[3] | CAPI/CASI/CAPI-by-phone/TBT (instruction only): Arabic, English, German, Kurmanji | Adolescents: 2,114 | Adolescents: 1,245 (778 with complete competence measurement) | 10/02/2020–05/09/2020 | Adolescents: 20 Euro drugstore voucher (post-paid) |

**Table 2** Methodological fact sheet for ReGES sample: Refugee Cohort 2.

*Note:* Data taken from several field reports (Ruland, Sandbrink, Cohrs, Weiß, & Hess, 2019; Ruland, Sandbrink, Cohrs, & Hess, 2019, 2020; Ruland, Sandbrink, Cohrs, Weiß, & Hess, 2019a, 2019b; Ruland, Steinwede et al., 2019) and Heinritz and Will (2021). The Response Rate 2 as well as the Cooperation Rate 4 were calculated according to AAPOR standards (The American Association for Public Opinion Research, 2016). This table is restricted to direct data collection with adolescents and parents. [1] Additonal data collected from educational staff in the (vocational) schools the adolescents attended at that time (PAPI); field phase deviating from adolescent/parent survey. [2] Additonal data collected from staff members in collective accommodation and municipalities (PAPI); field phase deviating from adolescent/parent survey. [3] The information on the duration of the interview refers to the pure interview time and includes neither the screening in Wave 1 nor the contacting phases. [4] The field phases of Waves 1 and 2 as well as Waves 4 and 5 overlap because after the personal interviews the respondents were invited to the online surveys by the interviewers; thus, the respondents were able to carry out these surveys relatively soon after the personal interview. [5] Due to the larger number of languages offered in the questionnaires via App, we included some additional cases that were not eligible for the waves with reduced languages. [6] Design switch due to the Corona pandemic (because household visits had to be cancelled, TBT was no longer possible; for details, see Will et al., 2020).

## 2.2 TIME OF DATA COLLECTION

ReGES aimed to gather detailed, close-knit information on the early phase of integration into the educational system. Seven waves of data collection were conducted between spring 2018 and autumn 2020 (for the field phases of the single data collection waves, see Table 1 for Refugee Cohort 1 and Table 2 for Refugee Cohort 2).

## 2.3 LOCATION OF DATA COLLECTION AND SAMPLING

ReGES sampled two cohorts of refugees within five German Federal States: Bavaria, Hamburg, North Rhine-Westphalia, Rhineland-Palatinate, and Saxony. These Federal States vary substantially in macrolevel structural characteristics such as the number of refugees allocated to them, their share of migrants, unemployment rate, and—last but not least—key characteristics of the educational system (esp. integration of recently arrived refugees in regular classes vs so-called newcomer classes designated primarily for language improvement). The limitation to five Federal States was made for cost-relevant and logistic reasons of data collection as well as reasons of content: It was a central aim of the ReGES study to include a sufficiently large number of refugees in the contexts under consideration, e.g. to enable analyzes of the influence of regional factors on educational success. In a Germany-wide survey, it would only have been possible to collect a sufficient number of cases in each Federal State if the total number of cases had been extremely high, due to the uneven distribution of refugees among the Federal States.

The sampling itself followed several steps: (1) selection of 40 cities and 80 communities (within 20 districts) on the basis of information from the Central Register of Foreigners, (2) sampling of individuals within these cities and communities via the respective Residents' Registration Offices,[1] and (3) screening of sampled individuals by pre-defined criteria (esp. refugee status) and asking for consent. Detailed information on the multi-step sampling procedure is given in Steinhauer et al. (2019). In subsequent waves, data collection also spread to other Federal States when respondents had moved home.

In order to attract refugee families to participate in the study, different strategies were applied: (1) Information events were held near all selected municipalities. At these events, stakeholders who deal with refugees in their everyday work were informed about the aims and procedures of the study, so that they were already aware of the study and could answer any questions the selected respondents might have. (2) The selected families were informed in detail about the study with the help of written material (cover letter, several specific flyers, data protection information). This material was given in German language as the spoken language at the refugees' homes was not available for study implementation. Via a QR code as well as a link to the project homepage, the respondents also had the opportunity to receive all necessary information about the study in their language of origin. In addition to German, seven other languages were offered: Arabic, English, Farsi, French, Kurmanji, Pashto, and Tigrinya. The selection of languages was intended to ensure that all respondents could be interviewed in at least one of the official languages of their country of origin (for details on language selection, see Gentile et al. 2019). (3) Respondents should be interviewed in their native language, if possible. In order to optimize the fit between interviewers and respondents, interview teams were deployed in all municipalities that could handle all of the eight survey languages offered. In addition, the interviewers had all information materials in all survey languages with them.

Since a relevant proportion of illiterates was assumed to be among the group of refugees who immigrated in the mid-2010s, audio files were implemented in the self-administered parts of the survey in Wave 1, so that people with little reading ability could also take part (see Gentile et al., 2019). However, due to the low use of audio files (see Heinritz et al., 2022), audio files were no longer used in the subsequent waves.

## 2.4 SAMPLE AND DATA COLLECTION

The ReGES samples of both Refugee Cohorts 1 and 2 included participants with a wide range of characteristics. Table 3 reports some basic descriptive information on parents and children in Refugee Cohort 1. Table 4 reports the respective information on parents and adolescents in Refugee Cohort 2.

All data collection was conducted by infas Institute for Applied Social Sciences, Bonn, Germany. Infas is a private social research institute with in-depth experience in scientific data collection—including longitudinal designs and complex measurement techniques.

## 2.5 MATERIALS/SURVEY INSTRUMENTS

One focus on the instrumentation was on a fine-grained assessment of the educational biography of refugee children and adolescents. Furthermore, information on their family situation, relevant aspects of their respective learning environments, personality and motivation facets, and migration/refugee-specific aspects was recorded. Detailed sociodemographic background variables as well as information concerning the flight history was included in the first wave.

A special feature of ReGES was the use of standardized test instruments: vocabulary was measured by the German version of the Peabody Picture Vocabulary Test. ReGES used the revised version PPVT-IV by Lenhard et al. (2015) with minor adaptations for the refugee population (see Obry et al., 2021). Grammar was measured by the German version of the Test for Reception of Grammar (TROG-D; Fox-Boyer, 2016). Within the TBT implementation of both receptive language tests, a number of (German) oral stimuli (word or sentence) was given to the respondent (via audio files) and the correct

| RELEVANT GROUP OF PEOPLE | VARIABLE | *M (SD)/ %* |
|---|---|---|
| Parent[1] | Sex of informant | |
| | *Male* | 78.1% |
| | *Female* | 21.9% |
| | Age of informant (in years) | 36.8 (6.7) |
| | Country of origin | |
| | *Afghanistan* | 9.0% |
| | *Iraq* | 13.2% |
| | *Syria* | 72.4% |
| | *Other* | 5.3% |
| | *Missing* | 0.1% |
| | Highest parental education (HISCED) | |
| | *No or primary education* | 45.7% |
| | *Secondary education* | 29.2% |
| | *Tertiary education* | 23.8% |
| | Missing | 1.3% |
| Child | Sex | |
| | *Male* | 52.3% |
| | *Female* | 47.7% |
| | Age at first interview | |
| | *4 years* | 19.5% |
| | *5 years* | 44.3% |
| | *6 years* | 27.7% |
| | *Older than 6 years* | 8.4% |
| | Care situation | |
| | *Preschool attendance* | 78.8% |
| | *Other types of extrafamilial care (exclusively)* | 1.5% |
| | *Home care* | 18.0% |
| | *Missing* | 1.8% |
| | Length of stay in Germany (in months) | 28.0 (9.0) |

**Table 3** Description of sample in Wave 1: Refugee Cohort 1.

*Note*: Percentages that do not add up to 100 are due to rounding. Source: doi:10.5157/ReGES:RC1:SUF:2.0.0. [1] Percentages of parental characteristics refer to parents who took part in the survey and not to the parental characteristics of the children and adolescents in the sample. Because some parents had multiple target children, these values can vary slightly. However, when parents had both Refugee Cohort 1 children and Refugee Cohort 2 children in the sample, their information was included in the description of both Refugee Cohort 1 and Refugee Cohort 2.

| RELEVANT GROUP OF PEOPLE | VARIABLE | *M (SD)/ %* |
|---|---|---|
| Parent[1] | Sex of informant | |
| | *Male* | 70.5% |
| | *Female* | 29.6% |
| | Age of informant (in years) | 45.7 (7.9) |
| | Country of origin | |
| | *Afghanistan* | 8.0% |
| | *Iraq* | 13.3% |
| | *Syria* | 73.7% |
| | *Other* | 4.7% |
| | *Missing* | 0.2% |
| | Highest parental education (HISCED) | |
| | *No or primary education* | 43.0% |
| | *Secondary education* | 30.0% |
| | *Tertiary education* | 26.4% |
| | Missing | 0.7% |
| Adolescent | Sex | |
| | *Male* | 55.1% |
| | *Female* | 44.9% |
| | Age | |
| | *14 years* | 14.1% |
| | *15 years* | 35.8% |
| | *16 years* | 31.8% |
| | *17 years* | 18.3% |
| | Educational situation: Type of school attended | |
| | *Hauptschule* (lower secondary track) | 19.8% |
| | *Realschule* (intermediate secondary track) | 22.0% |
| | *Gymnasium* (higher secondary track) | 21.9% |
| | *Gesamtschule* (integrates all tracks) | 16.8% |
| | *Verbundene Haupt- und Realschule* (combined lower and intermediate track) | 19.3% |
| | *Missing* | 0.2% |
| | Length of stay in Germany (in months) | 29.5 (9.1) |

**Table 4** Description of sample in Wave 1: Refugee Cohort 2.

*Note*: Percentages that do not add up to 100 are due to rounding. Source: doi:10.5157/ReGES:RC2:SUF:2.0.0. [1] Percentages of parental characteristics refer to parents who took part in the survey and not to the parental characteristics of the children and adolescents in the sample. Because some parents had multiple target children, these values can vary slightly. However, when parents had both Refugee Cohort 1 children and Refugee Cohort 2 children in the sample, their information was included in the description of both Refugee Cohort 1 and Refugee Cohort 2.

answer had to be picked out of four pictures for every oral stimulus on the tablet screen. Additionally, an indicator of basic cognitive functioning was available (Lang et al., 2014). The respondents worked on two item formats also given at a tablet: (1) Picture-Digit-Test: Based on a given list of stimulus-target combinations the respondents had to combine given stimuli to targets. (2) Matrices-Test:

Based on logical rules the respondent had to select the respective correct geometric forms in order to fill gaps in several given arrangements of geometrical forms. In all tests the instructions were translated.

More detailed information on the survey instruments for both Refugee Cohorts 1 and 2 is given in Table 5. At https://www.reges-data.de the complete instruments

| WAVE | INFORMANT | REFUGEE COHORT 1 | REFUGEE COHORT 2 | MODE |
|---|---|---|---|---|
| Wave 1 | Parents | Socio-economic background<br>Flight history<br>Accommodation history<br>Residence status<br>Migration-specific aspects<br>Refugee-specific aspects<br>Educational decisions<br>Returns to education<br>Educational biography of child<br>Personality of child<br>Motivation of child<br>Familial learning environment | Socio-economic background<br>Flight history<br>Accommodation history<br>Residence status<br>Migration-specific aspects<br>Refugee-specific aspects<br>Educational decisions<br>Returns to education | CAPI/CASI |
| | Children/ Adolescents | — | Migration-specific aspects<br>Refugee-specific aspects<br>Educational decisions<br>Educational biography<br>Personality<br>Motivation<br>Familial learning environment<br>Returns to education<br>Socio-economic background[1]<br>Flight history[1]<br>Accommodation history[1] | CASI |
| | Children/ Adolescents | Competence test:<br>German language (vocabulary and grammar)<br>Basic cognitive functioning | Competence test:<br>German language (vocabulary and grammar)<br>Basic cognitive functioning | TBT |
| | Educational staff | Institutional learning environment<br>Assessments of child[2] | Institutional learning environment<br>Assessments of adolescent[2] | PAPI |
| | Municipality staff | Regional context information | Regional context information | PAPI |
| | Staff in collective accommodation | Living context information | Living context information | PAPI |
| Wave 2 | Parents | Subjective perception of societal integration | Subjective perception of societal integration | App-based CAWI |
| | Children/ Adolescents | — | Subjective perception of societal integration | App-based CAWI |
| Wave 3 | Parents | Family context<br>Personality<br>Social capital<br>Educational placement of child | — | CATI |
| | Children/ Adolescents | — | Family context<br>Personality<br>Social capital<br>Educational placement | CATI |
| Wave 4 | Parents | Accommodation history (update)<br>Residence status (update)<br>Migration-specific aspects<br>Refugee-specific aspects<br>Educational decisions<br>Educational biography of child<br>Personality of child<br>Motivation of child<br>Familial learning environment<br>Returns to education | — | CAPI/CASI |
| | Children/ Adolescents | — | Accommodation history (update)<br>Residence status (update)<br>Migration-specific aspects<br>Refugee-specific aspects<br>Educational decisions<br>Educational biography<br>Personality<br>Motivation<br>Familial learning environment<br>Returns to education | CAPI/CASI |
| | Educational staff | Institutional learning environment<br>Assessments of child[2] | Institutional learning environment<br>Assessments of adolescent[2] | PAPI |

(Contd.)

| WAVE | INFORMANT | REFUGEE COHORT 1 | REFUGEE COHORT 2 | MODE |
|---|---|---|---|---|
| Wave 5 | Parents | Subjective perception of societal integration | Subjective perception of societal integration | App-based CAWI |
| | Children/ Adolescents | — | Subjective perception of societal integration | App-based CAWI |
| Wave 6 | Parents | Educational practices and values<br>Social desirability<br>Personality of child<br>Educational placement of child | — | CAWI |
| | Children/ Adolescents | — | Educational practices of parents<br>Educational values<br>Social desirability<br>Personality<br>Educational placement | CAWI |
| Wave 7 | Parents | Accommodation history (update)<br>Residence Status (update)<br>Migration-specific aspects<br>Refugee-specific aspects<br>Educational decisions<br>Educational biography of child<br>Personality of child<br>Motivation of child<br>Familial learning environment<br>Returns to education | — | CAPI/CASI |
| | Children/ Adolescents | — | Accommodation history (update)<br>Residence status (update)<br>Migration-specific aspects<br>Refugee-specific aspects<br>Educational decisions<br>Educational biography<br>Personality<br>Motivation<br>Familial learning environment<br>Returns to education | CAPI/CASI |
| | Children/ Adolescents | Competence test:<br>German language (vocabulary and grammar)<br>Basic cognitive functioning | Competence test:<br>German language (vocabulary and grammar)<br>Basic cognitive functioning | TBT |
| | Educational staff | Institutional learning environment<br>Assessments of child[2] | Institutional learning environment<br>Assessments of adolescent[2] | PAPI |

**Table 5** Content of the seven Waves for Refugee Cohorts 1 and 2.

*Note*: [1] Only in case the parents also do not take part in the survey. [2] For example, German skills or behaviour.

used (except competence tests) can be viewed in German and English (see block 'instrumentation' within the data documentation).

All survey instruments for the refugee families have been translated (with the exception of the competence tests as these are language-free or target to the measurement of German language competencies). The translation process in ReGES is based on the TRAPD model (see e.g. Survey Research Center, 2016) and is designed as a multi-stage process: translation, review, creation of a joint adjusted translation in the case of minor deviations and pre-tests in the case of serious deviations, and documentation (for more details on the translation process within ReGES, see Gentile et al., 2019).

## 2.6 QUALITY CONTROL
The development of the instruments and the data collection procedures was prepared at a series of expert meetings and underwent strict quality control measures such as checking the comprehensibility and cultural

appropriateness of the items used, quality control checks (especially concerning the translation of instruments), and intense interviewer training, supervision and feedback. Fieldwork checks included strict checks of fieldwork progress and interviewer performance (e.g. by real-life supervision of various personal interview settings and by examining recordings in CAPI and CATI surveys), selectivity checks, and data checks for missing values. Moreover, all data was intensely checked while editing the ReGES Scientific Use Files, and all data users were asked to give feedback to the LIfBi Research Data Center on possible errors in the datasets.

## 2.7 DATA ANONYMIZATION AND ETHICAL ISSUES
Data collection within ReGES was closely monitored by the LIfBi data protection team. They ensured that the data collection was based on a legally robust informed consent of the participants and followed the regulations of the European General Data Protection Regulation (GDPR). Of

utmost importance for achieving an informed consent was comprehensive information about the study design, the study aims, and the flow of data. Because data collection included questionnaires for school teachers, an approval process by the educational ministries of the five German Federal States under study was also needed (except in North Rhine-Westphalia where an information procedure is sufficient, and an active approval is not required).

Based on a detailed data protection concept, the data underwent strict anonymization procedures before being shared with the scientific community. Anonymization strategies are documented in the respective data manuals (FDZ-LIfBi, 2022a, 2022b). Depending on the respective sensitivity of the data, datasets are shared download, remote, or on-site (see Section 3).

### 2.8 EXISTING USE OF DATA

ReGES data is used by the ReGES team but is also open to the scientific community worldwide. All research projects that registered for ReGES data usage are listed at the ReGES data website (https://www.reges-data.de/en-us/Research/Projects). Moreover, all publications based on the ReGES data or related to the ReGES study are also listed at the ReGES data website (https://www.reges-data.de/en-us/Research/Publications). At the time of submission of this article, 16 registered projects and 28 publications are listed. The previous publication and research projects use the datasets of both Refugee Cohorts 1 and 2 and cover a wide range of topics.

## (3) DATASET DESCRIPTION AND ACCESS

The data collection in all seven ReGES survey waves is finished. All data underwent strict data checks, editing and anonymization procedures, as well as documentation routines. ReGES data is open for use to the scientific community worldwide and free of charge via the LIfBi Research Data Center following the FAIR principles (Findable, Accessible, Interoperable, Reusable; https://www.go-fair.org/fair-principles/). Beside the direct survey data, the datasets can be supplemented with regional variables. Using the option to add regional information to the ReGES data makes it possible to analyse educational processes while also taking regional characteristics into account (cf. Homuth et al., 2021, for an application example).

Depending on the sensitivity of the data, access is given: (1) as a download through the LIfBi Research Data Center, (2) via remote access, or (3) on-site within the secure environment. Datasets include raw data but also a set of generated variables (see Section 3.3). All data access is based on a valid contract.

### 3.1 REPOSITORY LOCATION

ReGES datasets are stored at the LIfBi Research Data Center (https://www.reges-data.de). The data—with a release on 08/12/2022—include all seven data collection waves (for Refugee Cohort 1: doi:10.5157/ReGES:RC1:SUF:3.0.0; for Refugee Cohort 2: doi:10.5157/ReGES:RC2:SUF:3.0.0).

### 3.2 OBJECT/FILE NAME

Data from Refugee Cohort 1 is delivered in the following files (listed are only download files in Stata; other dissemination versions[2] —such as the remote and on-site versions—as well as all SPSS files are organized in the same way):

- RC1_CohortProfile_D_3-0-0.dta
- RC1_ParentMethods_D_3-0-0.dta
- RC1_pChild_care_D_3-0-0.dta
- RC1_pParent_D_3-0-0.dta
- RC1_pTarget_D_3-0-0.dta
- RC1_pTargetCompetencies_D_3-0-0.dta
- RC1_spChildCare_D_3-0-0.dta
- RC1_spLanguageCourses_D_3-0-0.dta
- RC1_spParentAccomodation_D_3-0-0.dta
- RC1_TargetMethods_D_3-0-0.dta
- RC1_spParentSchool_duplicateEpisodes_D_3-0-0.dta

Data from Refugee Cohort 2 is delivered in a parallel way (again, only Stata download files are listed):[3]

- RC2_CohortProfile_D_3-0-0.dta
- RC2_ParentMethods_D_3-0-0.dta
- RC2_pParent_D_3-0-0.dta
- RC2_pTarget_D_3-0-0.dta
- RC2_pTargetCompetencies_D_3-0-0.dta
- RC2_spAccomodation_D_3-0-0.dta
- RC2_spEducation_D_3-0-0.dta
- RC2_spLanguageCourses_D_3-0-0.dta
- RC2_TargetMethods_D_3-0-0.dta

### 3.3 DATA TYPE

The ReGES datasets contain both raw data (after anonymization procedures) and processed data. Generated variables are offered in, for example, the following areas: country of origin of refugees, nationality, education of parents (ISCED), professional activity and professional status of parents in country of origin (e.g. ISEI, KLdB), professional activity and professional status of parents in Germany (e.g. ISEI, KLdB); and there is also generated data for easier use of the competence data (for the PPVT-IV see also Obry et al., 2021). Additionally, documents are available to help researchers make use of the data (see Section 3.9).

### 3.4 FORMAT NAMES AND VERSIONS

Datasets are delivered in SPSS and Stata. Additional documentation uses different formats such as text files, Excel files, and pdf files.

### 3.5 LANGUAGE

A comprehensive data manual is given in American English. A detailed codebook, the instruments used, as well as the datasets are provided in both German and American English. Field reports from the data collection institute are available in German only. All material is available at https://www.reges-data.de.

### 3.6 LICENSE

ReGES datasets are not deposited under an open license such as the Creative Commons Zero license. Instead, they are made available by the LIfBi Research Data Center based on a contract.

### 3.7 LIMITS TO SHARING

All datasets are made available without further delay after anonymization, documentation, and editing; there is no data embargo by the ReGES team. Data access is limited to researchers with an affiliation to a scientific institution and is for scientific purposes only. For data access, researchers have to sign a contract that especially regulates the scope and content of right of use (including concrete data recipients), data privacy, the processing of the personal data of the data recipient, and the obligation of the researchers to give feedback on publications based on the data. There are different contract versions for download, remote, or on-site access (https://www.reges-data.de/en-us/Data-and-Documentation/Data-Access).

### 3.8 PUBLICATION DATE

A first data version with data from Waves 1 and 2 was published 09/07/2021; the latest data version (with data from Waves 1, 2 and 3 of Refugee Cohorts 1 and 2) was delivered to the scientific community 10/01/2022.

### 3.9 FAIR DATA/CODEBOOK+

All ReGES datasets are made available to the scientific community following the FAIR guidelines via the LIfBi Research Data Center (https://www.reges-data.de/en-us/). The data documentation contains basic materials (data manual, release notes, data structure file, merging matrix), instruments (codebook, instruments), and fieldwork documentation (field reports). Access is given via: https://www.reges-data.de/en-us/Data-and-Documentation/Cohort-RC1 for Refugee Cohort 1 and https://www.reges-data.de/en-us/Data-and-Documentation/Cohort-RC2 for Refugee Cohort 2. To make data use more convenient, the ReGESplorer can be used to search for items or constructs used in the study (https://www.reges-data.de/en-us/Data-and-Documentation/Variable-Search). To support researchers, training sessions run by the LIfBi Research Data Center as well as e-mail and telephone support are available. Researchers can also use the LIfBi Research Data Center Forum (an open online discussion platform; mostly in German Language; https://forum.lifbi.de/).

## (4) REUSE POTENTIAL

ReGES data can be used to work on a variety of questions in the field of empirical educational research. Clearly following the guiding principles of life-course research (Elder & Giele, 2009; Elder et al., 2004) as well as the perspective of lifespan developmental psychology (Baltes, 1990; Baltes et al., 1980), the focus is on individual development as well as on transitions into and within the educational system and beyond. Educational trajectories and, more generally, life pathways of the refugee population that entered Germany in the mid-2010s can be described in a fine-grained fashion.

Exploiting the longitudinal structure, the data allows the identification of factors that are relevant not only for successful integration into the educational system but also for educational failure—defined by grades, certificates, competence status, and trajectory to subsequent educational institutions, or by broader indicators such as satisfaction and social integration. In addition, various issues can also be addressed in other areas such as migration research, developmental psychology, educational sciences, economy, and sociology of social inequality. Because educational research as well as other research areas often require longitudinal data, exploiting the full potential of secondary data analyses avoids not only the high costs of data collection (duplicated and therefore unnecessary) on the researchers' and funders' side but also any unnecessary strain on participants. Because ReGES data is shared by the LIfBi Research Data Center, every version of the datasets can be clearly cited and also used for both re-analyses and the analysis of completely new research questions. The very rich nature of ReGES data makes it impossible to process all potential research questions within the project team alone.

On the instrumentation level, an overlap to other studies—especially the German National Educational Panel Study (NEPS; Blossfeld & Roßbach, 2019) and the German Socio-Economic Panel (SOEP; Goebel et al., 2019)—was assured whenever possible. This opens up the option of comparing the ReGES refugees with representatively drawn (sub)samples of the population in Germany with comparatively little harmonization effort. It has to be kept in mind that the SOEP already includes a larger sample of refugees, and that the NEPS will include (due to changes in the population) a larger share of refugees in future cohorts or waves. But—even if other studies go along with larger sample sizes and especially contain more refugees in their samples)—the particular advantage of the ReGES study is the large number of refugee children and adolescents sampled in specific age groups at important transition points within the German educational system. This makes it possible to take into account also differences within the group of refugees (e.g. according to residence status, characteristics of school systems relevant for new immigrants, risk groups of post-

traumatic stress disorder) and to examine the effects of these differences on further educational trajectories.

One aspect of the ReGES data collection that is especially relevant for psychological research is the use of standardized tests of German vocabulary and grammar competencies as well as of basic cognitive functioning at two time points. Language competencies can be seen as a key for successful integration into (regular) school classes and as a determinant not only for educational and vocational success but also in terms of its fostering effects for integration into German society. Vocabulary is measured with a test that is widely used in the national as well as international context. This opens up the possibility of comparing different populations within Germany or refugee populations worldwide. Basic cognitive functioning was designed to serve as a relevant control variable.

Although the research questions that led to the design of the ReGES survey were clearly targeted on refugee children and adolescents, the data also contributes to a deeper understanding of children's and adolescents' contexts. There is a clear focus on the family—especially in Refugee Cohort 1—with detailed measures of the families' socio-economic background, their aspirations, and decision-making processes, as well as parenting behaviour, family climate, and home learning environment. Moreover, information from preschool and school teachers and principals could help answer research questions addressing the impact of institutional learning environments. Additionally, including information from staff members in collective accommodation and municipalities opens up research questions that clearly target the contextual embedding of refugee families in Germany.

Strengths of the data are the careful sampling process, the large sample size, the great willingness of the refugee families to cooperate, foreign language interviewing, the interdisciplinary instrumentation (including standardized competence tests), a high frequency of seven survey waves, and the connection to the instrumentation of other large surveys. Limitations of the ReGES data are the restriction to a selection of five Federal States, the restriction to a sample with quite secure residence status, attrition over time, and a mode effect resulting from the multi-method design.

Data collection will continue within the project 'Educational Trajectories of Refugee Children and Adolescents' funded by the German Federal Ministry of Education and Research. This will cover an observation period till 2024 and assess more data on educational transitions and integration into society. Because these measurements include a third competence assessment, this data will allow analyses of competence development. During this phase, two more CAPI interviews will be conducted and data will also be shared via the LIfBi Research Data Center.

## NOTES

1  Various criteria were specified for sampling at the individual level. In addition to the age groups of interest, only people were selected who arrived in Germany 2014 or later, have lived in Germany for at least three months and come from one of the main countries of origin of refugees who have good prospects of staying. Asylum seekers from countries with low prospects of staying (e.g. Balkan countries) were excluded from the ReGES sampling procedure. Overall, the ReGES data is not representative of the population of refugees in Germany (for a comparison with the data from the representative IAB-BAMF-SOEP survey of refugees see Will et al., 2021), but the data is well suited to examine covariations in the context of educational integration.

2  For remote and on-site use there are some additional data files on educational staff, collective accommodation, and municipalities: RC1_pEducator_care_R_3-0-0.dta; RC1_pInstitution_care_R_3-0-0.dta; RC1_xAccommodationStaff_R_3-0-0.dta; RC1_xMunicipalStaff_R_3-0-0.dta; RC1_pChild_school_R_3-0-0.dta; RC1_pEducator_school_R_3-0-0.dta; RC1_pInstitution_school_R_3-0-0.dta (remote version).

3  For remote and on-site use there are some additional data files on educational staff, collective accommodation, and municipalities: RC2_pEducator_R_3-0-0.dta; RC2_pInstitution_R_3-0-0.dta; RC2_xAccommodationStaff_R_3-0-0.dta; RC2_xMunicipalStaff_R_3-0-0.dta (remote version).

## ACKNOWLEDGEMENTS

We wish to express a special thanks to all the families, children, and adolescents who took part in our study. Moreover, we also thank the preschool and school principals and teachers, as well as the professionals on the local level who supported our data collection.

## FUNDING STATEMENT

The project ReGES was funded from July 2016 till December 2021 by the German Federal Ministry of Education and Research under Funding Number FLUCHT03. Data collection will continue within the project 'Educational Trajectories of Refugee Children and Adolescents', funded by the German Federal Ministry of Education and Research under Funding Number FLUCHT2021.

## COMPETING INTERESTS

The authors have no competing interests to declare.

## AUTHOR CONTRIBUTIONS

The study ReGES was located at the Leibniz Institute for Educational Trajectories (LIfBi). All departments of the LIfBi contributed to this project. Therefore, the authors thank not only the ReGES project team but also contributors from Department 1 ('Competencies, Personality, Learning Environments'), Department 2 ('Educational Decisions and

Processes, Migration, Returns to Education'), Department 3 ('Research Data Center, Methods Development'), the Center for Study Management, and also the LIfBi administration. More detailed information about the LIfBi is given at https://www.lifbi.de. Moreover, we thank the group of experts who helped to prepare the project on a more theoretical level. Data collection was conducted by the infas Institute for Applied Sciences. Our thanks go to the infas team and especially to the interviewers responsible for data collection in the field. More detailed information about infas is given at https://www.infas.de. The number of involved researchers, infrastructure experts and interviewers is so numerous that these can't be listed by names.

Concerning the paper both authors served within ReGES in leading positions, worked collaboratively on this paper and share joint responsibility for all its contents.

## AUTHOR AFFILIATIONS

**Jutta von Maurice**
Leibniz Institute for Educational Trajectories, DE
**Gisela Will** orcid.org/0000-0003-3249-4220
Leibniz Institute for Educational Trajectories, DE

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

## PEER REVIEW COMMENTS

*Journal of Open Psychology Data* has blind peer review, which is unblinded upon article acceptance. The editorial history of this article can be downloaded here:

- **PR File 1.** Peer Review History. DOI: https://doi.org/10.5334/jopd.77.pr1

von Maurice and Will *Journal of Open Psychology Data* DOI: 10.5334/jopd.77

**TO CITE THIS ARTICLE:**
von Maurice, J., & Will, G. (2023). Data from the Panel Study 'Refugees in the German Educational System (ReGES)'. *Journal of Open Psychology Data,* 11: 1, pp. 1–15. DOI: https://doi.org/10.5334/jopd.77

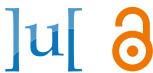