## [Peer Review History. · Journal of Open Psychology Data]

Comments to reviewer feedback

Paper Data from the Panel Study ‘Refugees in the German Educational System (ReGES)’

We gratefully thank the two anonymous reviewers for their helpful and constructive comments which contributed to a substantial improvement of our first paper version. Their feedback helped us improving the readability of the paper, avoiding unnecessary confusions and gave us valuable hints to missing information which we were happy to add in a revised version.

In the following table we answer the reviewer comments in detail.

Comments of reviewer A	Reaction to the comments of reviewer A
Figure 2 is appealing but challenging to understand, especially at that early point in the paper when the reader is not yet familiar with the study. I think two or three additional sentences with some instructions would be helpful for the reader to understand the content of the figure better.	Thanks for this comment concerning readability. The following paragraph has been added to better guide through Figure 2 within Section 2.1: Figure 2 presents the seven panel waves in ReGES. It shows primarily in which educational stages the refugee children and adolescents under study are ideally in at these seven waves and which central transitions in the German educational system they have to master during the progress of the panel. The upper half of Figure 2 focuses on Refugee Cohort 1, the lower half on Refugee Cohort 2. Figure 2 also shows in which waves data of children and adolescents (as the main targets of our research questions) is directly collected but also the measurement of additional information given from parents, educational professionals (preschool and school teachers and heads) and administration staff—following a clear multi-informant design.
The interviewing of refugee children, refugee adolescents, and their parents as target respondents is clear in the text. However, the reader learns rather incidentally that ReGES also interviewed educational and administrative staff. A clearer indication would contribute to a better understanding of the multi-informant design.	In the newly added paragraph in which we explain Figure 2 in more detail (see your first comment), we now also explicitly address the questioning of the context persons. We hope that this will contribute to a better understanding of the multi-informant design.
What is the technology-based aspect of the technology-based competence tests? Additional information on this would be interesting. Also, a reader may not be familiar with the test instruments described in section 2.5. I wonder how these tests (e.g., a Test of Reception of Grammar) are constructed to be used with	There is a lot of information that we could add about the competence measurements and its technological implementation. Due to length restrictions we decided to add some information in Section 2.5: Within the TBT implementation of both receptive language tests, a number of

children (at the age of 4+) who cannot read or write yet.	(German) oral stimuli (word or sentence) was given to the respondent (via audio files) and the correct answer had to be picked out of four pictures for every oral stimulus on the tablet screen. Additionally, an indicator of basic cognitive functioning was available (Lang et al., 2014). The respondents worked on two item formats also given at a tablet: (1) Picture-Digit-Test: Based on a given list of stimulus-target combinations the respondents had to combine given stimuli to targets. (2) Matrices-Test: Based on logical rules the respondent had to select the respective correct geometric forms in order to fill gaps in several given arrangements of geometrical forms. In all tests the instructions were translated. As you might have struggled with this point within Section 2.1, we included a hint to Section 2.5 there.
Table 1 / 2: Is the start time of Wave 2 on 29/01/2018 correct? Wave 1 has the same start time. There is also an overlap between the end of Wave 4 and the start of Wave 5.	Yes, this field phases overlap, but are still separable due to different mode and participant communication strategies. We added a note to Tables 1 and 2 (Section 2.1) explicitly stating this overlap and giving a reason for this overlap: The field phases of Waves 1 and 2 as well as Waves 4 and 5 overlap because after the personal interviews the respondents were invited to the online surveys by the interviewers; thus, the respondents were able to carry out these surveys relatively soon after the personal interview.
The relation between the realized sample in one wave and the gross sample in the next wave is somewhat confusing. For example, Table 1 shows 287 realized interviews with parents in Wave 2, but the size of the gross sample in Wave 3 increased to 1,935 parents. Were some respondents excluded from certain waves, or what is the reason? Can you provide more information on attrition across waves and respondent groups?	The gross sample was only changed substantially once, namely because from Wave 3 on only four languages were offered. Persons are only removed from the gross sample if they withdraw their willingness to participate in the panel, move abroad or were no longer able to participate due to long-term illness or death. Since the proportion of those who withdraw their willingness to participate in the panel is in the low single-digit range, the gross sample across all waves is still relatively large (see also the large gross sample in Wave 7). However, we see different responsiveness to the different survey modes. Within Section 2.1, we have added a paragraph to the manuscript and refer to further analyzes with the ReGES data that deal with the selectivity of the participants in the individual panel waves:

	As Table 1 and Table 2 show with respect to the valid realized sample participation rates vary substantially with the interview mode. While the parents and adolescents can best be reached in personal interviews (Waves 1, 4, and 7), followed by telephone interviews (Wave 3), online surveys (Waves, 2, 5, and 6) are the least effective. We also see different selectivity patterns, depending on the survey mode used (see Heinritz, & Will, 2021): The selective participation by education—which is particularly harmful for education-related studies—is least pronounced in the personal interview setting.
Sampling → Is there a rationale for sampling in only five federal states? → Did you restrict your sample to specific characteristics (e.g., certain countries of origin, date of arrival in Germany, command of the German language)?	Thanks for your detailed questions concerning the sampling procedure. As this was a very complex procedure, which we have documented in detail in other papers. But we were happy to add some additional information to the aspects you mentioned. Yes, there is a rationale for sampling only five Federal States. We added a paragraph on this in Section 2.3 of the manuscript: The limitation to five Federal States was made for cost-relevant and logistic reasons of data collection as well as reasons of content: It was a central aim of the ReGES study to include a sufficiently large number of refugees in the contexts under consideration, e.g. to enable analyzes of the influence of regional factors on educational success. In a Germany-wide survey, it would only have been possible to collect a sufficient number of cases in each Federal State if the total number of cases had been extremely high, due to the uneven distribution of refugees among the Federal States Yes, we included some sampling characteristics quite early in the process. We added a respective footnote within Section 2.3 regarding the restriction of the sample: Various criteria were specified for sampling at the individual level. In addition to the age groups of interest, only people were selected who arrived in Germany 2014 or later, have lived in Germany for at least three months and come from one of the main countries of origin of refugees who have good prospects of staying. Asylum seekers from countries with low prospects of staying (e.g. Balkan countries) were

→ How did you approach the selected refugees, inform them about the survey, and motivate them to participate (e.g., with translated letters of introduction or other material)?

excluded from the ReGES sampling procedure. Overall, the ReGES data is not representative of the population of refugees in Germany (for a comparison with the data from the representative IAB-BAMF-SOEP survey of refugees see Will et al., 2021), but the data is well suited to examine covariations in the context of educational integration.

The motivation of the refugees was highly important to start ReGES. We have added a paragraph in Section 2.3 describing the strategies we use to motivate respondents to participate in the study:

In order to attract refugee families to participate in the study, different strategies were applied: (1) Information events were held near all selected municipalities. At these events, stakeholders who deal with refugees in their everyday work were informed about the aims and procedures of the study, so that they were already aware of the study and could answer any questions the selected respondents might have. (2) The selected families were informed in detail about the study with the help of written material (cover letter, several specific flyers, data protection information). This material was given in German language as the spoken language at the refugees' homes was not available for study implementation. Via a QR code as well as a link to the project homepage, the respondents also had the opportunity to receive all necessary information about the study in their language of origin. In addition to German, seven other languages were offered: Arabic, English, Farsi, French, Kurmanji, Pashto, and Tigrinya. The selection of languages was intended to ensure that all respondents could be interviewed in at least one of the official languages of their country of origin (for details on language selection, see Gentile et al. 2019). (3) Respondents should be interviewed in their native language, if possible. In order to optimize the fit between interviewers and respondents, interview teams were deployed in all municipalities that could handle all of the eight survey languages offered. In addition, the interviewers had all information

	materials in all survey languages with them.
Translating the survey instruments into various languages was certainly not a trivial undertaking. It would be beneficial to have some information about the translation process of the instruments and the decision into which languages instruments were translated.	Here we have added further information in two different places. First, we added information on the languages used in Section 2.3 (see comment above). Second, in Section 2.5 we included a paragraph on the translation process and especially gave the hint to a publication from the ReGES team targeting on the translation process: All survey instruments for the refugee families have been translated (with the exception of the competence tests as these are language-free or target to the measurement of German language competencies). The translation process in ReGES is based on the TRAPD model (see e.g. Survey Research Center, 2016) and is designed as a multi-stage process: translation, review, creation of a joint adjusted translation in the case of minor deviations and pre-tests in the case of serious deviations, and documentation (for more details on the translation process within ReGES, see Gentile et al., 2019).
Table 5 contains the content of seven waves in general terms (e.g., migration-specific aspects, refugee-specific aspects). It is understandable that a detailed description of all instruments would go beyond the scope of this paper, but references to the documentation of the questionnaires would be perfect.	We provide detailed information on our data Website and therefore added a sentence to this when introducing Table 5 within Section 2.5: More detailed information on the survey instruments for both Refugee Cohorts 1 and 2 is given in Table 5. At https://www.reges-data.de the complete instruments used (except competence tests) can be viewed in German and English (see block 'instrumentation' within the data documentation).
Section 3.5: Can you provide references, links, or information on the location where the user can easily find the codebooks, questionnaires, and field reports?	All these information can easily be found on our data Website. We therefore added the following sentence in Section 3.5: All material is available at https://www.reges-data.de
Contribution statement: According to the Paper Template, authors were asked to “list all contributions towards this manuscript, including the contributions of all individuals who helped to collect the data (who may also not be an author of the data paper), including their roles and affiliations at the time of data collection.” The authors of previous JOPD articles indicated their responsibilities in the study and/or their contributions to the paper. While the author(s) of this article acknowledge the contributions of several departments of the	As several hundred people contributed to our study we are unfortunately not able to list them by names. We make this clearer in our contribution statement: The number of involved researchers, infrastructure experts and interviewers is so numerous that these can't be listed by names. Moreover, we gave a short statement concerning the contributions of both authors to the paper: Concerning the paper both authors served

LifBi without naming specific persons, I would assume that the journal also would expect this section to describe each author's contribution to the paper. However, this information is most likely currently missing because this is a blinded document.	within ReGES in leading positions, worked collaboratively on this paper and share joint responsibility for all its contents.
Comments of reviewer B	Reaction to the comments of reviewer B
In my view, it would be helpful for understanding the scope of the different waves better if tables 1 and 2 included an indicator on the length of the questionnaire, e.g. mean/medium interview duration.	Thank you for this important note, we have added the interview durations (with arithmetic mean M and standard deviation SD to Tables 1 and 2 within Section 2.1 for all panel waves).
Particularly in studies on migrants and especially on refugees, questions of selectivity and thus of generalizability and validity play a major role. Therefore, I would like to suggest some further statements on this topic in the text. This concerns in particular the process of sampling (Can the data be considered representative for all refugees in Germany? Do the distributions in the data differ from the population? Are weights available?) ... ... and the implementation of the survey (Were multilingual interviewers available? Were there audio files for illiterate persons?).	You have raised several important aspects which we were happy to react to. We added some more information on the sampling criteria and representativeness of the data in Section 2.3 (footnote): Various criteria were specified for sampling at the individual level. In addition to the age groups of interest, only people were selected who arrived in Germany 2014 or later, have lived in Germany for at least three months and come from one of the main countries of origin of refugees who have good prospects of staying. Asylum seekers from countries with low prospects of staying (e.g. Balkan countries) were excluded from the ReGES sampling procedure. Overall, the ReGES data is not representative of the population of refugees in Germany (for a comparison with the data from the representative IAB-BAMF-SOEP survey of refugees see Will et al., 2021), but the data is well suited to examine covariations in the context of educational integration. Picking up your question, we added some further information in Section 2.3: In addition to German, seven other languages were offered: Arabic, English, Farsi, French, Kurmanji, Pashto, and Tigrinya. The selection of languages was intended to ensure that all respondents could be interviewed in at least one of the official languages of their country of origin (for details on language selection, see Gentile et al. 2019). In addition, we have added a paragraph on the use of audio files also in Section 2.3, addressing your question of illiterate persons: Since a relevant proportion of illiterates was assumed to be among the group of refugees who immigrated in the mid-2010s, audio

	files were implemented in the self-administered parts of the survey in Wave 1, so that people with little reading ability could also take part (see Gentile et al., 2019). However, due to the low use of audio files (see Heinritz et al., 2022), audio files were no longer used in the subsequent waves.
In sections 3.1 and 3.8, statements are made about which waves are already available. While one gets the impression in the methods-section that all seven waves are already available because their data collection was completed two years ago, one learns in section 3.1 that only the first three waves are available. It would be helpful if the reader were given this information at an earlier stage and learned the reasons for the delay in making the data available. In addition, the second sentence under 3.1 ("first waves ... are already accessible") seems to be wrong and should be checked.	The original formulation was correct when submitting the first paper version. We are very happy that we were able to release data of all panel waves on 8th of December. We updated the information accordingly in Section 3.1: The data—with a release on 08/12/2022—include all seven data collection waves (for Refugee Cohort 1: doi:10.5157/ReGES:RC1:SUF:3.0.0; for Refugee Cohort 2: doi:10.5157/ReGES:RC2:SUF:3.0.0) We also updated the number of registered projects, the file names as well as references concerning data documentation to that latest (and complete) data version.
The reuse-section provides specific and useful suggestions. The authors also mention other relevant data sets in the field of refugee studies. However, I would suggest to consider including some more information on the (existing) added value of the ReGES data compared to these other (representative) surveys.	We have added a paragraph on the special analysis potential of the ReGES data in Section 4: But—even if other studies go along with larger sample sizes and especially contain more refugees in their samples—the particular advantage of the ReGES study is the large number of refugee children and adolescents sampled in specific age groups at important transition points within the German educational system. This makes it possible to take into account also differences within the group of refugees (e.g. according to residence status, characteristics of school systems relevant for new immigrants, risk groups of post-traumatic stress disorder) and to examine the effects of these differences on further educational trajectories.